# Registering Unmanned Aerial Vehicle Videos in the Long Term

**DOI:** 10.3390/s21020513

**Published:** 2021-01-13

**Authors:** Pierre Lemaire, Carlos Fernando Crispim-Junior, Lionel Robinault, Laure Tougne

**Affiliations:** 1Université Lumière - Lyon 2, LIRIS, UMR5205, F-69676 Lyon, France; laure.tougne@liris.cnrs.fr; 2Foxstream, F-69120 Vaulx-en-Velin, France; l.robinault@foxstream.fr

**Keywords:** registration, stabilization, unmanned aerial vehicle, drone

## Abstract

Unmanned aerial vehicles (UAVs) have become a very popular way of acquiring video within contexts such as remote data acquisition or surveillance. Unfortunately, their viewpoint is often unstable, which tends to impact the automatic processing of their video flux negatively. To counteract the effects of an inconsistent viewpoint, two video processing strategies are classically adopted, namely registration and stabilization, which tend to be affected by distinct issues, namely jitter and drifting. Following our prior work, we suggest that the motion estimators used in both situations can be modeled to take into account their inherent errors. By acknowledging that drifting and jittery errors are of a different nature, we propose a combination that is able to limit their influence and build a hybrid solution for jitter-free video registration. In this work, however, our modeling was restricted to 2D-rigid transforms, which are rather limited in the case of airborne videos. In the present paper, we extend and refine the theoretical ground of our previous work. This addition allows us to show how to practically adapt our previous work to perspective transforms, which our study shows to be much more accurate for this problem. A lightweight implementation enables us to automatically register stationary UAV videos in real time. Our evaluation includes traffic surveillance recordings of up to 2 h and shows the potential of the proposed approach when paired with background subtraction tasks.

## 1. Introduction

Unmanned aerial vehicles (UAVs) have gained popularity within the past few years as a means of acquiring data in a number of tasks [1,2]. In particular, tethered drones [3,4,5] have traded mobility and a short battery lifespan for the ability to sustain several hours of stationary flight, at heights up to 80 m above ground. The industry advertises them as temporary video surveillance cameras (https://elistair.com/, https://tethereddronesystems.co.uk/, https://www.ntpdrone.com/), to which their video flux is indeed quite comparable. However, their viewpoint is not constant over the course of a video. Using off-the-shelf background subtraction (BGS), comparing object trajectories on such flux is, at best, tedious (Figure 1, left column in particular). Traditionally, registration and stabilization are two distinct classes of solutions which have been proposed to address this type of issue. In the case of stationary drone videos, both approaches answer a similar problem with distinctive advantages and drawbacks, amongst which are jitter and drifting. Jitter refers to the video being subject to fast, low-magnitude movements. The video is then described as shaky. Drifting is the tendency to slowly change viewpoint over time, sometimes with high magnitude. Jitter and drifting may be due to the acquisition itself, even despite the use of a gimbal; however, they may also be linked to flaws in the post-processing registration or stabilization techniques. Such flaws tend to appear in lengthy sequences and/or in the presence of numerous mobile objects. However, both characteristics are not often featured within available datasets, and few studies have focused on them. This is detrimental to typical real-world applications, in particular for cases such as traffic monitoring or crowd surveillance in which stationary UAVs could be very useful [2].

In [6], we proposed a hybrid method that was reported to handle jitter and drifting well over sequences of up to 15 min, featuring many mobile objects. We proposed the modeling of motion estimation methods, assimilated to matrix products, by decomposing them into two terms: an optimum and an error. Then, we showed how to combine registration and stabilization into a hybrid solution that is jitter-free and driftless. However, this approach was only applied to 2D-rigid transforms, which is a rather limited transformation space for this problem. In this paper, we generalize the theoretical ground of this approach and show how to expand it to perspective transforms, or 2D homographies, which require space-specific adjustments. Our study shows that upgrading from 2D-rigid transforms to perspective transforms alone significantly improves the performance of the application. By conducting an evaluation on video sequences that last up to two hours, sometimes even while using a gimbal, we demonstrate that this hybrid method has a very positive impact on further processing such as off-the-shelf background subtraction.

## 2. Prior Work

In our work, video registration must be understood as the action of keeping the background content of the video stable over the course of a video. Our application purpose includes traffic surveillance from stationary UAVs, where many objects may enter and leave the field of view. Therefore, we cannot expect the existence of a dense mapping between every pixel from one video frame to one another, including foreground pixels, as in the case of the registration hypothesized in [7].

Our problem is to estimate an image transformation that optimally compensates for camera motion. When a significant part of the image (i.e., its background) is available throughout the video sequence, we can estimate this transformation between any current frame and a reference image. This is the video registration approach. A stabilization approach does not usually make such an assumption. In this scenario, the camera may change angle and even content over the course of a sequence, and the goal is to remove shakiness, as is often the case with hand-held devices. Under such circumstances, the image content, including the background, is not necessarily consistent over time. The typical approach can be understood, very roughly, as primarily estimating a trajectory, which is defined as a combination of short-term inter-frame motions. This trajectory is then filtered to generate a smooth trajectory. Finally, the image is re-projected to follow the desired, stable trajectory [8,9].

Other techniques are designed to determine the camera position and orientation, such as Structure from Motion (SfM) [10] or Simultaneous Localization and Mapping (SLAM) [11]. Unfortunately, they tend to be confused by the relative lack of parallax from background objects and the motion of numerous foreground objects.

In the case of a stationary UAV, we may assume that a background is constantly visible over the course of a sequence. Therefore, this results in a situation in which both registration and stabilization techniques apply and should ideally provide similar results.

The straightforward application of registration techniques has been reported to generate a high-frequency noise, referred to as jitter, in previous studies [12,13,14]. From our understanding, this is related to the typical implementation of registration methods. Most are based on a sparse feature points matching process, and then the rejection of outliers through robust estimators such as RANdom SAmple Consensus (RANSAC). While this is computationally very effective, thresholds in the matching of feature points, or in the inliers selection process, tend to cause some points to be sporadically selected. This can cause jitter. This issue may be tackled by estimating the motion on a more global (holistic) scale, as in [15] or more recently by using deep-learning, regression-based approaches, such as in [16] or [17]. Holistic approaches tend to be computationally expensive as compared to feature-point approaches and still do not prevent the occurrence of noise, as theorized in Section 3.1. Another approach is to add temporal coherence into the solution, which is closer to stabilization techniques, as in the present work.

Standard stabilization techniques [8,9] have been applied to drone videos. In [18], the authors were able to simultaneously stitch and stabilize video from a swarm of drones. Sparse feature points matching, such as the Kanade Lukas Tracker (KLT) [19], is often used to estimate inter-frame motion with various strategies: one is to track several points during a video sequence, thus producing a number of tracklets that are assembled to estimate global motion; another is to directly estimate the frame-to-frame motion given a transformation space. Specific optical flow models have also been proposed to favor spatial coherence [20]. Convolutional Neural Networks have been proposed, such as in [21], where the authors sought to reproduce the behavior of gimbals by using a specifically designed device to which two cameras were attached, with one of them physically stabilized by a gimbal to produce the ground-truth. While appealing, this approach has proved impractical for our problem [6]. In general, approaches which do not hypothesize the existence of constant background should not be expected to stay registered to a constant background.

Most of the previously cited works about camera motion compensation state that this problem is made increasingly difficult by lighting and content changes. These phenomena are almost impossible to avoid with longer video sequences, especially in scenarios such as traffic or crowd monitoring which feature numerous mobile objects. In our case, the use of tethered drones, which allows virtually unlimited flight time, has emphasized the need to address those issues. Based on our observations, stabilization techniques have shown limitations when registering video sequences longer than several minutes. Despite progress in designing efficient and robust motion estimators, such as in [22], computationally light registration techniques can become noisy over time. However, errors in both situations are very distinctive when compared to a theoretical, noise-free registration.

In [6], we proposed to take advantage of the difference in the behavior of motion estimators between stabilization and registration contexts to propose a hybrid solution. We have used the computationally inexpensive KLT approach for both registration and stabilization parts. This simple scheme allowed for the real-time, online handling of long sequences featuring multiple objects. However, the presented model applied only to simple, 2D-rigid transforms, the expressiveness of which, according to the present evaluation, is limited. In this paper, we rework the proposed model and show how to apply this approach to more relevant homographies. We provide an evaluation which shows the benefits of our approach for challenging videos in an applied context.

## 3. Modeling the Problem

### 3.1. Image Warping

In both cases, stabilizing or registering a video is the action of compensating for undesired camera motion while preserving the image content variability over time.

This compensation is performed using an image alignment; that is, a warping operation applied to an image so that it matches the view of another image.

Let $H\subset R^{3}$ be the space of homogeneous image coordinates. Defining a warping transformation to register from an image to another is relevant under the assumption that a significant part of both images is shared. Classically, warping methods such as Thin Plate Splines (TPS) [23], homographies or rigid transforms [24] are bijections when in non-degraded cases. As such, we define W:H→H as the space of bijective warping transforms.

Let $w_{1}$, $w_{2}$, and $w_{3}$ be elements of W, the space of warping transformations. It is possible to compose warping transformations. The composition of $w_{1}$ first and then $w_{2}$ is denoted as
$$w_{3}=w_{2}\circ w_{1}$$

We can also define the inverse transform $w^{-1}$ as
$$Id=w\circ w^{-1}$$
where w∈W and Id is the identity (the coordinates in H were unchanged during the transform). Under those conditions, W is a group.

Suppose that we want to register two images $p_{i}$ and $p_{j}$ of a similar scene but taken with a different camera pose. We denote w(i,j)∈W as the warping transform determined to warp from image $p_{i}$ to image $p_{j}$. Note that
$$w{\left(i,j\right)}^{-1}=w\left(j,i\right).$$

#### 3.1.1. Video Registration

Let us align every image in the video sequence to a reference frame denoted as $p_{0}$. The straightforward application of a motion estimation solution to register a video (Figure 2) may be expressed as
(1)$$w_{r}\left(i\right)={\tilde{w}}_{r}\left(i,0\right)$$
where $w_{r}\left(i\right)$ is the applied warping transform at frame *i* and $\tilde{w_{r}}$ is the motion estimator used for the registration task.

One issue is that performing frame-to-frame alignment in this fashion does not take into account temporal coherence. Most camera movements are compensated, but an additional high-frequency noise, referred to as jitter by the authors [12,13,14], tends to be introduced by the solution.

#### 3.1.2. Video Stabilization

Jitter in video is avoided by the use of stabilization techniques.

Stabilizing a video is typically a problem of smoothing the observed camera movements over the course of a video. Although not always explicitly described this way in the literature, we can model it by constructing a short-term trajectory; by a short-term trajectory, we mean a series of transforms that is representative of recent motion relative to the background in the video. We define a trajectory as a series of warping transforms:(2)$$\left\{\begin{array}{c} t_{\tilde{w_{s}}}\left(0\right)=Id\\ t_{\tilde{w_{s}}}\left(i\right)=\tilde{w_{s}}\left(i-1,i\right)\circ \delta \left(i\right)\circ t_{\tilde{w_{s}}}\left(i-1\right)\;\mathrm{with}\;i>0 \end{array}\right.$$
where $t_{\tilde{w_{s}}}\left(i\right)\in W$ the trajectory position at time *i*, $\tilde{w_{s}}$ the motion estimator used for the stabilization task, and δ(i)∈W a decay term, when it is needed. It should be close or equal to Id further details about this term are provided in Section 5.2.

We define $T_{\tilde{w_{s}}}^{n}=\left\{t_{\tilde{w_{s}}}\left(i\right);0\leq i\leq n\right\}\in T_{W}$ as a trajectory of length n+1 within $T_{W}$. $T_{W}$ is the set of trajectories in W.

A stabilization solution aims at removing the high-frequency motion, considered as noise, from the trajectory. It requires the definition of a filter that is able to smooth trajectories.

This kind of filter is a function $F:(T_{W},N)\to W$ that produces a warping transform at a given frame from a trajectory. We denote this kind of filter as $F(T_{\tilde{w_{s}}}^{n},i)$ with i≤n the index of the image to which we apply the filter.

In an online context, the filter can only use trajectory elements from the previous and the current frames, meaning that n=i.

To stabilize the video at frame *i*, we first cancel the observed trajectory $t_{\tilde{w_{s}}}\left(i\right)$, then replace it by the corresponding smoothed trajectory $F(T_{\tilde{w_{s}}}^{n},i)$ (Figure 3):(3)$$w_{s}\left(i\right)=F\left(T_{\tilde{w_{s}}}^{n},i\right)\circ t_{\tilde{w_{s}}}{\left(i\right)}^{-1}$$
with n≥i.

While not designed to register to a constant background over the course of a video, this approach is effectively designed to remove high-frequency motion relative to the background.

Note that when δ(i) is the identity, calculating $T_{\tilde{w_{s}}}^{n}$ becomes a method of measuring motion from the initial point in the video. Therefore, it may seem acceptable to use it to register the video; however, this measure tends to slowly accumulate error, which is a form of drifting. We elaborate on this phenomenon in the following section.

## 4. Proposed Approach

In this section, we adapt the method presented in [6] to the more general formalism presented in this paper. We propose to model the motion estimation problem through the following decomposition:(4)$$\tilde{w}\left(i,j\right)=E_{\tilde{w}}\left(i,j\right)\circ w^{\mathrm{opt}}\left(i,j\right)$$
where $\tilde{w}\left(i,j\right)\in W$ is the estimated camera motion between $p_{i}$ and $p_{j}$, $w^{\mathrm{opt}}\left(i,j\right)\in W$ corresponds to the unique optimal transform from $p_{i}$ to $p_{j}$. It is measured as background motion between $p_{i}$ and $p_{j}$, and $E_{\tilde{w}}\left(i,j\right)\in W$ corresponds to a motion estimation error.

The general idea is to assert the existence of a unique transform $w^{\mathrm{opt}}\left(i,j\right)\in W$ that compensates optimally for the camera movements between $p_{i}$ and $p_{j}$ while preserving their content, given the studied warping space.

A range of registration methods, either holistic [15] or sparse [19], is available in the literature. They all aim at estimating $w^{\mathrm{opt}}\left(i,j\right)$ independently of inter-frame variations (illumination changes, mobile objects, acquisition noise, etc.). In other terms, their goal is to minimize the magnitude of the term $E_{\tilde{w}}\left(i,j\right)$ given their respective constraints and application domains. Minimizing this term is important to achieve good performances. However, we argue that this kind of error cannot be completely avoided.

Following Equations (Equation 1) and (Equation 4), we have
(5)$$w_{r}\left(i\right)=w^{\mathrm{opt}}{\left(0,i\right)}^{-1}\circ E_{{\tilde{w}}_{r}}{\left(0,i\right)}^{-1}$$

Following Equations (Equation 2) and (Equation 4), we have
(6)$$t_{\tilde{w_{s}}}\left(i\right)=E_{{\tilde{w}}_{s}}\left(i-1,i\right)\circ w^{\mathrm{opt}}\left(i-1,i\right)\circ \delta \left(i\right)\circ t_{\tilde{w_{s}}}\left(i-1\right)$$

Assuming that camera movements are such that a constant background is consistently available over the course of the video, we have, by construction,
(7)$$w^{\mathrm{opt}}\left(0,i\right)=w^{\mathrm{opt}}\left(i-1,i\right)\circ \ldots \circ w^{\mathrm{opt}}\left(0,1\right)$$
which allows us to define an equivalent error relative to the initial position, for each element in $T_{\tilde{w_{s}}}$:(8)$$t_{\tilde{w_{s}}}\left(i\right)=E_{\tilde{w_{s}}}^{\mathrm{equiv}}\left(0,i\right)\circ w^{\mathrm{opt}}\left(0,i\right)$$

For instance,
(9)$$t_{\tilde{w_{s}}}\left(1\right)=E_{\tilde{w_{s}}}\left(0,1\right)\circ w^{\mathrm{opt}}\left(0,1\right)\circ \delta \left(1\right)$$
Thus, we have
(10)$$E_{\tilde{w_{s}}}^{\mathrm{equiv}}\left(0,1\right)=E_{\tilde{w_{s}}}\left(0,1\right)\circ w^{\mathrm{opt}}\left(0,1\right)\circ \delta {\left(1\right)}^{-1}\circ w^{\mathrm{opt}}{\left(0,1\right)}^{-1}$$

When δ(1) is close to Id, $E_{\tilde{w_{s}}}^{\mathrm{equiv}}\left(0,1\right)$ is close to $E_{\tilde{w_{s}}}\left(0,1\right)$. It is possible to construct $E_{\tilde{w_{s}}}^{\mathrm{equiv}}\left(0,i\right)$ in a similar manner. This equivalent error term becomes a composition between successive error terms $E_{\tilde{w_{s}}}\left(i-1,i\right)$, decay factors and the optimal motion estimation.

The attractiveness of this model lies in the fact that the nature of the terms $E_{{\tilde{w}}_{r}}\left(0,i\right)$ and $E_{\tilde{w_{s}}}^{\mathrm{equiv}}\left(0,i\right)$ is different.

The more dissimilar $p_{i}$ and $p_{j}$, the more significant $E_{\tilde{w}}\left(i,j\right)$ is likely to be. With consecutive images being rather similar, their error term is likely of a low magnitude. However, in such cases, foreground objects often exhibit little movement. This motion then tends to be wrongly considered as background motion by the motion estimator, which contributes in part to the term $E_{\tilde{w}}\left(i,j\right)$. We believe this to be a generic behavior and that it is non-specific to the motion estimation solution chosen in a particular case.

Designing a robust registration or stabilization technique in the presence of a constant background is therefore equivalent to estimating $E_{\tilde{w_{r}}}\left(0,i\right)$ or $E_{\tilde{w_{s}}}^{\mathrm{equiv}}\left(0,i\right)$. While we cannot measure this directly, we can take advantage of a combination of both Equations (Equation 5) and (Equation 6).
(11)$$\begin{array}{cc} & d\left(i\right)=t_{\tilde{w_{s}}}\left(i\right)\circ {\tilde{w}}_{r}\left(i,0\right)\\ T_{d}^{n}= & \left\{d(i);0\leq i\leq n\right\}\in T_{W} \end{array}$$

Following the notation in Section 3.1.2, we have
$$d\left(i\right)=E_{\tilde{w_{s}}}^{\mathrm{equiv}}\left(0,i\right)\circ E_{\tilde{w_{r}}}{\left(0,i\right)}^{-1}$$

We have stated that $E_{\tilde{w_{s}}}^{\mathrm{equiv}}\left(0,i\right)$ should be a low-frequency term, referred to as drifting, and $E_{\tilde{w_{r}}}\left(0,i\right)$ should be a high-frequency, noisy term, which is referred to as jitter. Under those conditions, applying a filter such as the one discussed in Section 3.1.2 allows us to isolate one error term from another.
(12)$$F\left(T_{d}^{n},i\right)\approx E_{\tilde{w_{s}}}^{\mathrm{equiv}}\left(0,i\right)$$

Finally, the proposed registration is as follows:(13)$$w_{p}\left(i\right)=F\left(T_{d}^{n},i\right)\circ t_{\tilde{w_{s}}}{\left(i\right)}^{-1}$$

An overview of the final process is proposed in Figure 4.

In a similar manner to [6], the construction here shows that, as long as the error term $E_{\tilde{w_{s}}}^{\mathrm{equiv}}\left(0,i\right)$ is of a low frequency, this combination holds. This implies that the trajectory can be built in the short term, which practically allows the use of more refined warping spaces as homographies (Section 4.2), as in the original paper.

### 4.1. Application to 2D-Rigid Transforms

In the previous section, we proposed a hybrid combination of registration and stabilization techniques that is designed to provide a jitter-free registration solution. This model holds under the strict condition that motion estimators and all the compositions stay within the group W.

In the case of 2D-rigid transforms, the W space is better represented by a quartet of parameters $\left(t_{x},t_{y},\alpha ,s\right)$ from which we can classically determine the 3 × 3 matrices used for 2D linear transforms in the homogeneous space [24]. The composition can be computed directly by using the quartet $\left(t_{x},t_{y},\alpha ,s\right)$. Staying in W is guaranteed as long as s≠0.

Under those conditions, the decay factor δ(i) can be neutral, meaning that it is equal to Id. This is the configuration presented in [6]. As proposed in the latter, a Kalman Filter (KF) [25] is applied to those $\left(t_{x},t_{y},\alpha ,s\right)$ parameters independently, allowing online operation.

### 4.2. Application to Perspective Transforms

In our context, the UAV is stationary and films towards the ground, and its height is significantly higher than the captured vertical objects. In this case, perspective transforms (homographies) are a much more accurate model than 2D-rigid transforms. They are also represented by 3 × 3 matrices, and they are defined up to a scale factor. However, multiplying such matrices does not guarantee that we can stay within the group of homographies. Interpolation (needed for filtering) should also not be performed directly on 3 × 3 matrices.

One can represent homographies by associating two sets of ordered quartets of 2D points. We used this to compose and interpolate between homographies. Every element of W is represented by the displacement it causes on the boundaries of an image of fixed size (w,h), with its dimensions corresponding to those of our video frames. The four corners of the image and their respective positions after the perspective transform can be described with eight variables (2D coordinates for four points), allowing us to generate a unique, eight-DOF (degree of freedom) homography matrix. Following the works of Hartley and Zisserman [24], degraded cases occur when the resulting tetragon (in our representation, the coordinates of the image corners after the homography has been applied) contains colinear points. In this case, defining an inverse transform is no longer possible, and we do not stay within the group W.

We now need to define how to compose and interpolate between homographies. Composition is performed as follows: let $h_{a}$ and $h_{b}$ be two homographies. Composing homographies $h_{b}$ and $h_{a}$ is performed by calculating the image corner positions after warping them the first time according to $h_{a}$ and then a second time according to $h_{b}$. Solving between the original image corner positions and the resulting corner positions provides us with a new $h_{c}$ homography, which is the result of $h_{b}\circ h_{a}$.

Interpolation is performed by applying a unique coefficient to the displacement of all four corners. Interpolating between homographies implies that every intermediate position between two homographies stays within W. Avoiding degraded cases implies that this representation must be restricted strictly to convex sets of ordered quartets of 2D points. We did not find this restriction to cause any drawback. From our understanding, concave (diabolo-shaped) or triangular sets of points should not occur when registering real-life images within our context.

To lower the risks of falling into a degraded case, we need to prevent the series of transforms from diverging infinitely. For this reason, we need the term δ in (Equation 2) to be non-neutral. Through δ, in a small proportion, we integrate the term F(d,i) into the calculation of $t_{\tilde{w_{s}}}{\left(i\right)}^{-1}$. This proposition only holds under the assumption that our solution is accurate enough. It can be understood as a positive feedback loop.

Note that representing homographies as a set of corner displacements is only used to allow operations such as inverting, composing and interpolating between elements of W. Motion estimators such as $w_{r}$ and $w_{s}$ are still computed with the help of standard methods such as KLT, where many feature points are matched from one image to another, and least square adjustment and/or robust estimators can be applied.

## 5. Evaluation

### 5.1. Evaluation Dataset

To evaluate our method, we used 10 RGB, 1080p (Full HD) video sequences, of which the shortest eight had already been used in [6]. Most sequences were acquired at 25 fps, in daylight, and featured multiple mobile vehicles. Four of them, namely “C0”, “C1”, “C2” and “C4”, were filmed from overhead with a GoPro camera attached to a light drone pointed directly towards the ground along the vertical (yaw) axis. As was unknown to us, the mounting system of the camera seemed to offer little stabilization, especially when compared to the later-described L2 sequence. The videos are not shaky, but feature occasionally significant camera motion. Videos “C1” and “C4” are 4 min long and display few vehicles and only light motion. “C2” is also 4 min long, but features heavy motion with two fast 180° clockwise yaw motions performed by the human operator (Figure 5). The 15 s long “C0” is an excerpt from “C1” in which all vehicles are static, which was produced with the aim of evaluating the proposed methods in the absence of foreground motion. “MFull” is a one-hour long video, acquired with a CMOS camera which was attached without a gimbal to a tethered drone. The drone was positioned approximately 50 m above ground in stationary flight next to a congested roundabout. Overall, the sequence features heavy viewpoint motion and continuous foreground motion from multiple vehicles. “MFull” was cut into four successive subsequences of 15 min in length, namely “M1”, “M2”, “M3” and “M4”, which were already featured in [6]. Note that heavy motion is exhibited during “M4”, while the UAV was lowered by a few meters during “M3”, producing a significant change in the aspect of the scene. None of the previously mentioned sequences displayed significant rolling shutter issues. Finally, “L2” is a 120 min long contiguous sequence streamed from a tethered drone. The camera was a Yangda sky-eye-30Hz 1080p 30× zoom camera (only light zoom was used), which is a camera mounted on a gimbal specifically designed for drones, making this sequence very stable overall. The video displays light rolling shutter, compression artifacts and lighting changes but few mobile objects. Figure 1 provides examples from the sequences “MFull” and “L2”. This shows the high variety of camera poses and mobile vehicles in MFull, and the high variety of lightning conditions during L2.

### 5.2. Implementation

We used the OpenCV version 4.3 library [26] and C++ to implement the approaches evaluated in this paper. Regarding 2D-rigid transforms, the settings were identical to those presented in [6].

In both 2D-rigid and perspective cases, images were set to one-channel grayscale and resized to 576 × 24. Motion estimators $\tilde{w_{s}}$ and $\tilde{w_{r}}$ were performed through the KLT approach. In both cases, we extracted 200 Shi–Tomasi corners [19] and used the Lukas and Kanade Pyramidal Optical Flow (LKPOF) algorithm [27] as a tracking solution.

For 2D-rigid transforms, all tracked points were used to compute $\tilde{w_{s}}$ following the least square adjustment. For $\tilde{w_{r}}$, only the best 100 points according to their matching score in the LKPOF algorithm were used, as a simple step to reject the worst outliers. For unclear reasons, our experiments with RANSAC and its OpenCV implementation were unsuccessful with 2D-rigid transforms. Motion estimation matrices were converted to the quartet $\left(t_{x},t_{y},\alpha ,s\right)$ mentioned in Section 4.1, and composition was performed directly on the quartet. Those four parameters were fed to a Kalman Filter (KF) with the scale parameter *s* converted into a logarithmic scale, meaning that the KF worked linearly with this parameter.

For perspective transforms, we used all tracked points (at most 200 following the LKPOF method) and the robust estimators implemented in OpenCV to generate homography matrices. Consecutive transforms were implemented by using the non-parametric least median algorithm, while the registration transforms were implemented by using RANSAC, with the error and threshold parameters set as default following the OpenCV implementation. The reason why we used different estimators between both motion estimations is that the KLT approach performed on consecutive images at 30 fps produces few outliers, allowing the least median robust estimator to be efficient while computationally less expensive. Meanwhile, the estimator used to compute $\tilde{w_{r}}$ used the classical RANSAC procedure on the 100 best-matched points according to the LKPOF algorithm, since outliers were likely to occur. Following the homography matrix estimations, we determined the corresponding displacement of corners of the image as specified in Section 4.2 to allow us to compose and interpolate between transforms. Filtering was performed through a KF applied to the corners set of the image, in the Euclidean space.

In both cases, our implementation used CPU-only operation and ran at a minimum of 60 fps, on a 2.5 Ghz Intel Core i7 MacBook Pro. This enabled real-time operation and left room for further processing analysis.

None of our experiments seemed to indicate a particular sensitivity towards a given set of parameters.

### 5.3. Evaluation Protocol

To show the benefits of the proposed combination, we have compared it with different arrangements of its elementary components, in both 2D-rigid transforms and homography configurations. The following settings were evaluated:*Raw*: the original, unprocessed video.*StabKalRig* (or *StabKalPer*): the video stabilized by the algorithm described in Equation (Equation 3), using the same computation of $\tilde{w_{s}}$ and filter as described in Section 5.2, in the 2D-rigid (or the 2D homographies) domain.*RegLPosRig* (or *RegLPosPer*): the video registered by the algorithm described as $\tilde{w_{r}}$ in Section 5.2, with the registration proposed at frame i−1 as an initialization for the registration of frame *i*, in the 2D-rigid (or the 2D homographies) domain.*RegKalRigid* (or *RegKalPer*): the video registered by the algorithm described as $\tilde{w_{r}}$ in Section 5.2, with a KF set as described in Section 5.2 for both the initialization and the filtering, in the 2D-rigid (or the 2D homographies) domain.*CombiRig* (or *CombiPer*): the proposed combination method, as described in Section 4, in the 2D-rigid (or the 2D homographies) domain.

It is necessary to quantify two different properties: the registration (the ability of a solution to remain registered to the same viewpoint), and the stability (the ability of a solution to avoid background motion). Lacking the ground-truth for camera movements and the image content, we used the same protocol as in [6] as a basis and added another measurement.

Registration properties were evaluated with the frame displacement (*fd*) measure, which consisted of measuring the median displacement of a set of feature points tracked from the reference frame to the current frame. In practice, we tracked 200 Shi–Tomasi corners through LKPOF, following the KLT approach. To reject the worst outliers, the measure was performed using only the 100 best points according to their matching score.

Stabilization properties were evaluated through a measure called the mean pixel values difference (*mpvd*), which consisted of measuring the evolution of pixels’ grayscale intensity between consecutive images. In practice, we warped frames according to the evaluated setting; then, we calculated the absolute difference between consecutive frames, pixel by pixel, thus forming a new image. The *mpvd* measure consisted of computing the average pixel value on this differential image, only within the area where both warped consecutive images were overlapped. This measure was computed on 8-bit images converted to floating points, and thus ranged from 0 to 255. Our assumption was as follows: when the video is stable, only mobile objects should cause pixel values to change significantly from one frame to the next one; when the video is unstable, the intensity values of a significant part of the pixels from the background may change in addition to foreground objects, causing this measure to increase.

In general, better registration (or stability) was found with the lowest *fd* (or *mpvd*) measure.

In addition to the evaluation protocol available in [6], we proposed a last performance measurement related to the actual application field of the proposed approach. This consisted of applying a baseline background subtraction model, without any form of smoothing or filtering, to the warped video. The ratio of segmented foreground pixels, denoted as *fgr*, fused both stability and registration properties. When the video was correctly registered—i.e., the background was stable—only pixels related to foreground motion were classified as foreground. When the background drifted or was shaky, pixels from the background may also have been wrongly classified as foreground. Thus, we believe that, in general, the lower *fgr* is, the better. This measure shows the relevance of different registration or stabilization approaches in an applied context. For this measure, and because of its computational efficiency, we used ViBe [28] without any form of filtering or morphology transform. However, since ViBe, as well as other background subtraction models, is reported to handle shaky videos well (meaning that it integrates jitter into its background model), *fgr* was only used complementary to *mpvd* to measure stability properties.

### 5.4. Results

First, we must assess how the tested methods remained registered to an initial viewpoint. The results in Table 1 show the mean *fd* measure for all tested videos. As anticipated, stabilization approaches (StabKalRig and StabKalPer) performed poorly. All of the tested 2D-rigid registration techniques (RegLPosRig, RegKalRig, CombiRig) displayed comparable results on relatively short sequences with little camera motion, such as “C0”, “C1” and “M2”.

However, the *fd* measurement was globally lower on registration methods in the perspective domain, such as RegLPosPer, RegKalPer and foremost the proposed combination of CombiPer. For our dataset, 2D-rigid transforms were not expressive enough to compensate for camera motions. In both cases, a filtered registration (RegKalRig, RegKalPer) suffered from the inertia and sometimes even strong divergence (RegKalPer on M4 and C2, for instance), that occurred when the optical flow algorithm was badly initialized, and it could not track enough points to provide a good registration. It is also important to note that the use of a gimbal, as in L2, did not provide viewpoint registration. With the *fd* indicator, results for the L2 raw video were remarkably similar to those obtained with software stabilization. On all tested videos, when the proposed method could not provide the best performance, it was well within a subpixel range from the best solution. The same can be stated when only using rigid transforms, as in [6], even with the longer sequences L2 and MFull. This is emphasized in Figure 6 and Figure 7, which show the evolution of the *fd* indicator on the longer sequences of the database. On those figures, we can observe the significantly improved stability of the proposed method over time. As shown by a linear regression on the various curves displayed, the proposed combination in the perspective domain was remarkably stable, although following a slightly increasing slope. The trend is however much more stable than other tested approaches, which the results suggest to be more affected by noise. Figure 6 and Figure 7 also show that 2D-rigid transforms may have reached an optimum, while much improvement can be observed with the use of perspective transforms.

Table 2 shows the stability properties of tested approaches according to the *mpvd* indicator. Results emphasize the effect of jitter, with registration solutions (both in rigid and perspective cases) performing even more poorly than the original video on longer sequences, including MFull and L2. Filtered registration techniques (RegKalRig, RegKalPer), even when diverging, provided better stability performances. Meanwhile, stabilization techniques displayed significantly better performance, with a slight overall advantage towards homographies. Results for L2 show that even a video acquired from camera mounted on a gimbal in a stationary flight can be positively affected by software stabilization. The proposed combination, especially when using perspectives, was the most consistent solution of all, suggesting that it is jitter-free and driftless.

The last proposed evaluation showed the impact of our approach in an applied context. Table 3 displays the *fgr* using a baseline background subtraction algorithm, ViBe [28]. ViBe parameters were kept as default, as provided by the authors. In essence, static backgrounds are supposed to give better results, which tends to favor registration solutions over drifting videos. ViBe is also designed to handle shaky images, to a certain extent.

The results show that perspective transforms suited this problem consistently better than other options. However, even while using only rigid transforms, the proposed combination showed decent performance, while the proposed combination using perspective transforms resulted in a significantly lower amount of segmented foreground pixels than all other solutions. More interestingly, the one hour long MFull sequence displayed a similar *fgr* to its 15 min subsets M1, M2, M3 and M4, while the other competing registration methods tended to output a higher ratio for the full sequence than for its subsets, suggesting that jitter tends to increase over time (see also Figure 8). Our study shows that the proposed combination could remain consistently registered without adding jitter, even when handling long sequences. This behavior was emphasized on the two hours long sequence L2, where CombiPer performed better in the long term than other approaches (Figure 9). Analyzing the trends in Figure 8 and Figure 9 with the help of linear regressions suggests that the proposed method in the perspective domain resulted in very stable results for both videos, which seems rational given their content (consistency in terms of foreground content and drone motion). On the other hand, for a given approach, the behaviors differed from one video to another. For L2, a growing slope was linked to growing noise over time (Figure 9, RegLPosPer). For MFull, the difference in performances between CombiPer and RegLPosPer seemed to be less linked to a general trend and more to changes in the drone viewpoint than for L2. The presence of a gimbal on L2 may explain the behavioral difference between both situations.

For a better overview of the results provided by the *fgr* measure, we have included in Figure 10 and Figure 11 examples from the sequence MFull where we applied ViBe to the output of each method. At the time of the test-frame, the drone was unintentionally lowered by a few meters. A crop is included to show precisely how the background was preserved between the various tested algorithms and to see the output of the BGS segmentation more clearly. Figures show that tested stabilization solutions did not preserve a constant background alignment, while registration techniques tended to perform better. However, significant differences can be seen regarding the behavior of ViBe. In both 2D-rigid and perspective domains, we can notice noise in the BGS output, which was caused by jitter. This noise is particularly noticeable around edges in the background, such as trees and building contours. Filtering the output of the registration algorithm actually makes this worse, because it generates inertia and prevents the solution from staying well registered even on average. Meanwhile, our proposed combination allowed the BGS output to be less noisy; in particular, in the perspective domain where almost no static object is segmented. It can be observed that the road topology was well preserved on the CombiPer line (Figure 11), while vertical objects such as buildings were stretched because of the change in viewpoint.

## 6. Conclusions and Perspectives

This paper addresses the problem of generating a constant viewpoint from videos acquired by stationary UAVs. Through the theoretical study of motion estimation models, we show how to combine a short-term trajectory and a long-term registration technique into a registration method that is jitter-free and driftless. We show how to apply this to perspective transforms that are well suited to UAV imagery. The use of low-cost, simple bricks allows for real-time application, as a pre-processing step, on lengthy sequences featuring a significant number of mobile objects.

Our study shows that this approach can be efficiently applied to improve the use of off-the-shelf background subtraction solutions on stationary UAV-acquired videos. We also show that such an approach can effectively handle videos with or without the use of physical stabilization devices such as gimbals. Depending on the circumstances, it may be possible to rely on this kind of computational solution rather than an expensive physical stabilization device. However, an end-user should keep in mind that gimbals or optical stabilization devices can help with other issues such as vibrations, which may induce blur or rolling shutter issues not addressed in this work. Using our approach may also allow the use of captive balloons as an environmentally friendly, cheaper replacement to stationary drones, despite their tendency to be heavily affected by windy conditions. We would be very interested in trying our approach on such devices.

In our implementation, a Kalman Filter (KF) was used to smooth what can be understood as the difference between a short-term trajectory and a long-term registration. After the short-term trajectory was updated with a new frame, the KF allowed us to predict where to initialize the next registration estimation. Since a motion estimation step is essentially a minimization problem, a better initialization means a more efficient and accurate estimation. Another advantage of our method is its ability to handle the temporary disappearance of the background provided in the reference frame (for instance, when very large camera movements occur, or when foreground objects hide most of the background). When the registration estimation is unable to converge, we can set it as the identity of the proposed combination. Under this circumstance, our approach is still capable of staying registered to the initial viewpoint by relying on the short-term trajectory inversion. One drawback of the proposed hybrid combination is that we always assume that the short-term trajectory estimation produces a low-magnitude, slowly evolving error term. If the short-term trajectory happened to generate a high-magnitude, high-frequency noise, our method would be likely to fail, since it would be unable to differentiate the error terms caused by registration and stabilization, respectively. One way to avoid this is to work with high fps video acquisition. The more similar consecutive images are, the less likely erroneous short-term trajectory estimations will be. For real-time application, this may imply that the end-user has to find a trade-off between frame-rate and resolution.

Although our evaluation shows that we can properly handle video sequences of up to two hours, significantly longer sequences remain a challenge that still needs to be addressed. In our approach, we can compensate noise in the registration step by filtering out its corresponding motion estimation. This step relies on the assumption that this estimation is, although very noisy, averagely accurate. This assumption may not hold when drastic appearance changes occur; for instance, when we try to register a night view from a scene that was initialized in daylight. More sophisticated motion estimation techniques can be integrated in the general scheme of our approach and may prove helpful to tackle this kind of challenging situation. Another approach might be, in this case, to design a reference frame-updating procedure. In future works, we will focus on this challenging situation.

## Figures and Tables

**Figure 1 sensors-21-00513-f001:**
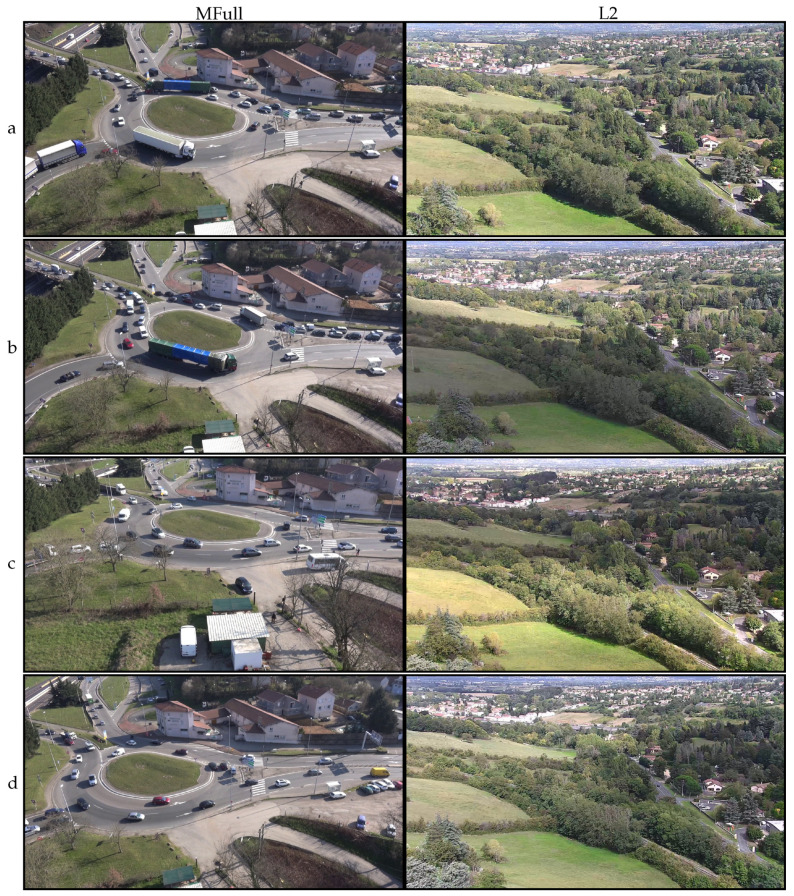
Images extracted from the one hour-long MFull (**left column**) and the two hours-long L2 (**right column**) sequences in our database, described in Section 5.1. (**a**): reference frame. (**b**,**c**): intermediate frames. (**d**): frames picked near the end of the sequence.

**Figure 2 sensors-21-00513-f002:**
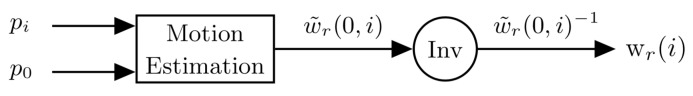
Typical registration procedure. “Inv” corresponds to an inversion in the space of warping transforms.

**Figure 3 sensors-21-00513-f003:**
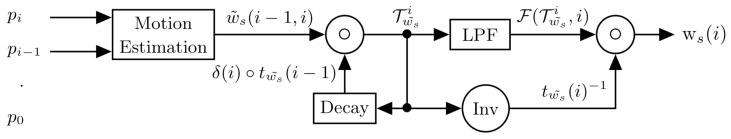
Typical stabilization procedure. “Inv” corresponds to an inversion in the space of warping transforms; LPF stands for low-pass filter.

**Figure 4 sensors-21-00513-f004:**
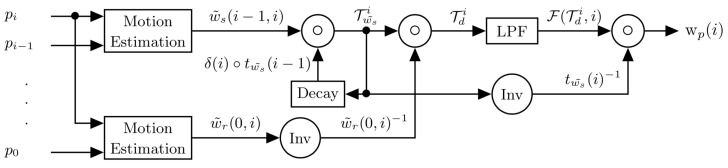
Overview of the proposed method. “Inv” corresponds to an inversion in the space of warping transforms; LPF stands for low-pass filter.

**Figure 5 sensors-21-00513-f005:**
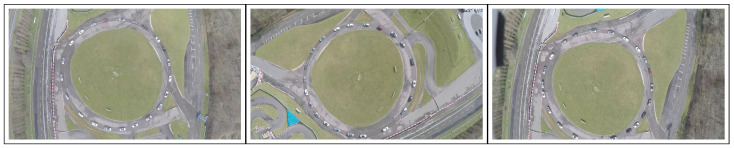
Images extracted from the five-minute long C2 sequences, taken approximately 30 s apart. The left image is the reference frame.

**Figure 6 sensors-21-00513-f006:**
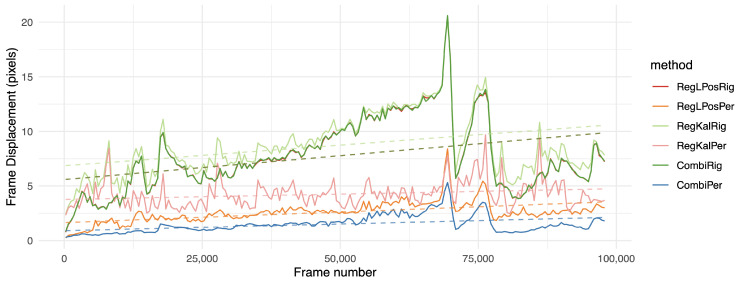
Frame displacement (*fd*) indicator over time on the MFull sequence; the lower the position on the graph, the better. Values were aggregated by averaging on intervals of approximately 500 frames. For more clarity, we left out the less relevant methods. The dashed line shows the result of a linear regression over the whole sequence.

**Figure 7 sensors-21-00513-f007:**
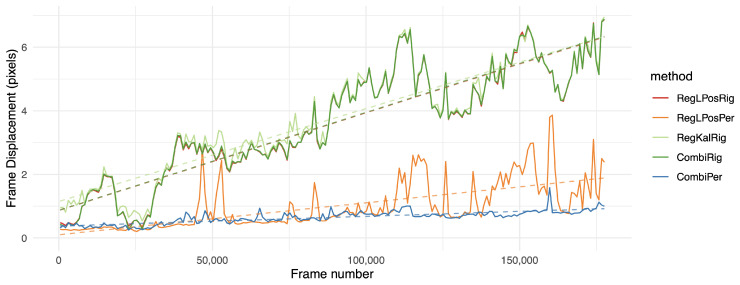
Frame displacement (*fd*) indicator over time on the two hours long L2 sequence. The lower the position on the graph, the better. Values were aggregated by averaging on intervals of approximately 1000 frames. For more clarity, we left out the less relevant methods. The dashed line shows the result of a linear regression over the whole sequence.

**Figure 8 sensors-21-00513-f008:**
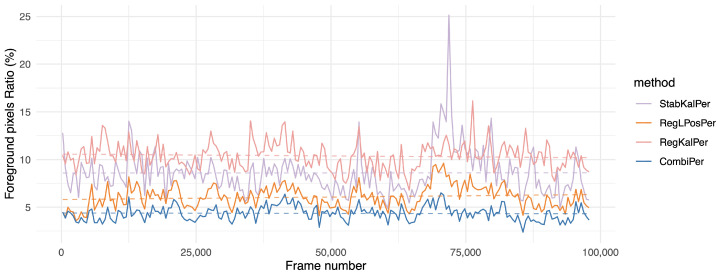
Evolution of the foreground pixel ratio (*fgr*) indicator over time for the MFull sequence (the lower, the better). Values were aggregated by averaging on intervals of approximately 500 frames. For more clarity, we kept only the methods using homographies. The dashed line shows the result of a linear regression over the whole sequence.

**Figure 9 sensors-21-00513-f009:**
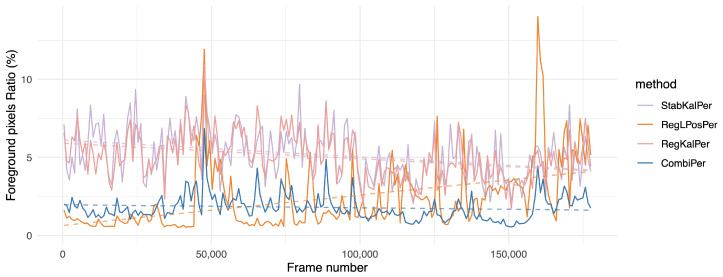
Evolution of the foreground pixel ratio (*fgr*) indicator over time on the two hours long L2 sequence (the lower, the better). Values were aggregated by averaging on intervals of approximately 1000 frames. For more clarity, we kept only the methods using homographies. The dashed line corresponds to a linear regression over the whole sequence.

**Figure 10 sensors-21-00513-f010:**
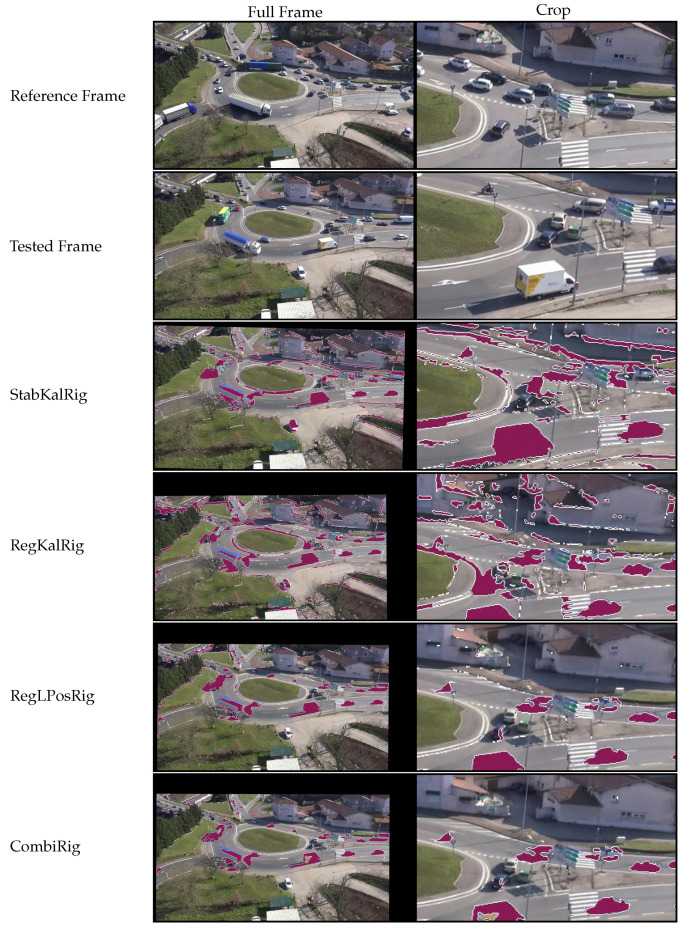
Results from tested 2D-rigid methods on the sequence MFull. The test-frame was picked approximately 38 min after the start of the video. To emphasize differences, we have displayed the output of the background subtraction algorithm ViBe [28] on tested configurations (segmented parts in purple, contours in white). The full frame (**left column**) is cropped at a fixed location (**right column**) to show details more clearly.

**Figure 11 sensors-21-00513-f011:**
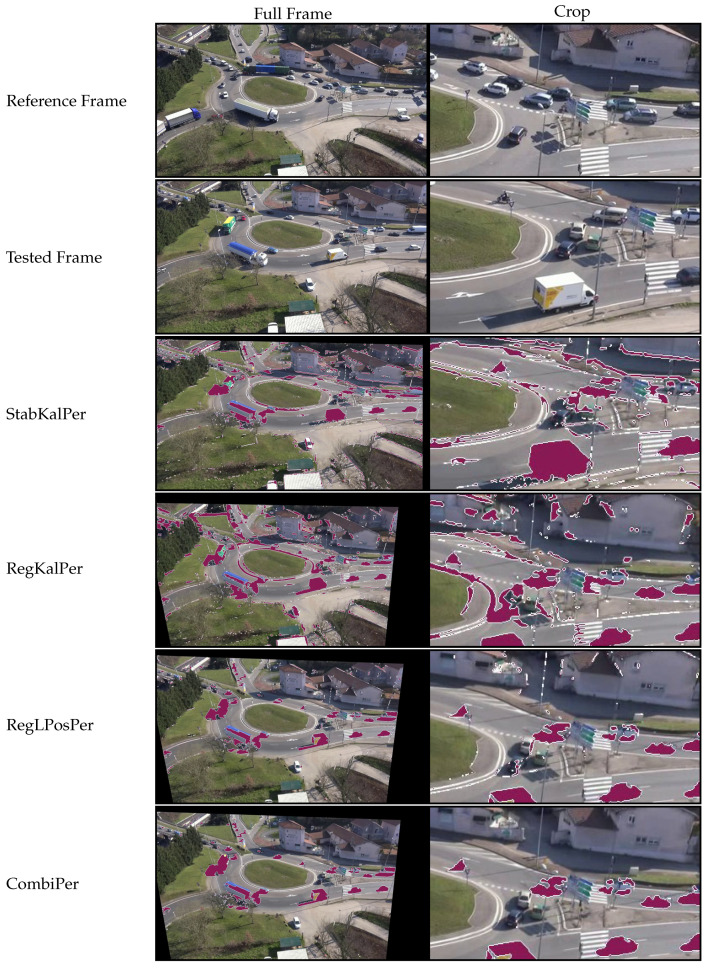
Results from the tested perspective methods on the sequence MFull. The test-frame was picked approximately 38 min after the start of the video. To emphasize differences, we have displayed the output of the background subtraction algorithm ViBe [28] on tested configurations (segmented parts in purple, contours in white). The full frame (**left column**) is cropped at a fixed location (**right column**) to show details more clearly.

**Table 1 sensors-21-00513-t001:** Mean frame displacement (*fd*) on our test sequences (pixels). **First rank**; second rank; third rank.

Settings	M1	M2	M3	M4	MFull	C0	C1	C2	C4	L2
Raw	17.54	17.98	14.9	27.35	19.53	10.39	11.94	20.17	9.19	12.83
StabKalRig	18.95	17.85	15.28	26.41	19.79	12.4	13.47	20.37	10.5	12.92
StabKalPer	17.73	17.82	14.53	27.42	19.31	12.38	13.07	18.45	10.66	12.91
RegLPosRig	4.9	3.86	6.55	11.94	7.73	0.81	1.33	4.73	1.1	3.6
RegLPosPer	1.6	0.89	1.74	3.0	2.6	**0.44**	**0.68**	1.41	**0.8**	1.0
RegKalRig	6.26	5.47	7.39	12.59	8.71	1.04	1.49	14.07	1.3	3.75
RegKalPer	3.74	4.04	3.7	22.42	4.25	0.8	0.87	17.59	1.0	7.03
CombiRig	4.91	3.9	6.59	11.96	7.74	0.81	1.34	4.74	1.13	3.61
CombiPer	**0.8**	**0.71**	**1.13**	**2.05**	**1.51**	0.6	0.72	**1.02**	0.9	**0.65**

**Table 2 sensors-21-00513-t002:** Mean pixel value difference (*mpvd*) between consecutive frames (grayscale value). **First rank**; Second rank; Third rank. Higher values tend to be caused by jitter while lower values show the stability properties.

Settings	M1	M2	M3	M4	MFull	C0	C1	C2	C4	L2
Raw	6.81	7.2	7.15	7.25	6.78	2.03	2.2	2.71	2.26	3.27
StabKalRig	3.24	3.32	3.38	3.3	3.18	1.51	1.6	2.09	1.65	2.38
StabKalPer	3.08	3.08	3.15	3.12	2.96	1.47	1.58	2.32	1.63	**2.32**
RegLPosRig	6.46	4.96	6.54	7.88	7.42	2.32	2.39	3.92	2.11	6.77
RegLPosPer	6.6	4.69	6.35	6.52	7.52	2.22	2.64	4.49	2.26	6.78
RegKalRig	5.37	5.6	5.61	5.8	5.49	1.51	1.65	2.21	1.67	2.74
RegKalPer	5.33	5.58	5.53	6.92	5.36	1.5	1.65	2.35	1.68	2.48
CombiRig	3.24	3.25	3.38	3.41	3.27	1.44	1.6	1.59	1.62	2.5
CombiPer	**2.95**	**3.01**	**3.06**	**3.04**	**2.9**	**1.41**	**1.57**	**1.51**	**1.61**	2.37

**Table 3 sensors-21-00513-t003:** Ratio of segmented foreground pixels (*fgr*) within the database as a percentage. **First rank**; Second rank; Third rank. Lower values suggest that less background motion was wrongly considered as foreground motion.

Settings	M1	M2	M3	M4	MFull	C0	C1	C2	C4	L2
Raw	15.03	14.87	14.22	14.75	14.53	5.19	3.52	4.89	3.46	11.04
StabKalRig	10.46	10.01	9.99	10.01	9.91	4.61	2.14	4.16	2.23	5.16
StabKalPer	9.04	8.61	8.83	9.41	8.6	4.42	2.01	4.65	2.13	5.16
RegLPosRig	6.87	6.11	7.1	8.5	7.77	0.4	1.0	2.52	1.08	3.15
RegLPosPer	5.84	4.76	5.61	5.5	6.07	**0.15**	0.84	2.05	**1.01**	2.38
RegKalRig	10.94	11.18	10.62	11.68	11.23	0.98	1.28	4.42	1.41	4.84
RegKalPer	10.5	10.87	9.77	13.77	10.38	0.63	1.1	4.2	1.37	5.02
CombiRig	6.51	6.07	6.67	7.83	7.27	0.39	0.97	2.37	1.09	2.82
CombiPer	**4.28**	**4.45**	**4.39**	**4.47**	**4.36**	0.26	**0.83**	**1.53**	1.12	**1.8**

## Data Availability

The data presented in this study are available on request from the corresponding author. The data are not publicly available due to yet unsolved questions regarding the drivers privacy within the presented videos.

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
