# Peer review of "Registering Unmanned Aerial Vehicle Videos in the Long Term"

_sensors, 2021, doi:10.3390/s21020513_

Round 1
Reviewer 1 Report
This study proposes a method to generate real-time constant viewpoint from videos acquired by stationary UAVs. Its main contributions consist in combining short-term trajectory with a long-term registration technique resulting in a registration method which is jitter-free and driftless.
The document is easy to read and follow.
The English is in general clear but should be revised.
The subject of the paper is interesting, with potential of application.
Authors should add images to illustrate de results and not only graphs so that readers can have a clearer understanding of the impact of the proposed method in the final results since you claim jitter-free and driftless.
For the sake of comparison authors should also consider including in their acquisition setups a gimbal to compare the effects of video with and without gimbal.
The conclusions section is very limited. Authors should detail more their analysis.
Although authors refer in the final paragraph of conclusions section that in the future the proposed method should cope with lighting variations during day long videos the problem is that even in short time videos that light variation can occur which is very common. This is one of the drawbacks of the proposed method.
In line 20 Fig.1 is referred but only appears 2 pages after?! Please correct.
The caption of Figure 1 seems too long. The description of the videos at least should be written inside a paragraph.
The first reference to Fig. 2 in the text appears after de image insertion. Please correct.
Table 1, Figure 5, 6 captions should also be reduced and the commenting part relocated to a paragraph.
In line 320 Table 3 is referenced after presenting the table. Please correct.
In line 343 Fig. 8 is referred but appear in the middle of references?! Please correct.
Author Response
Please see the attachment.
It includes a letter common to both reviewers as well as the updated manuscript (starting at page 8). We hope that this way of answering your review is convenient.

Reviewer 2 Report
General considerations
The paper shows a workflow to reduce the jitter and drifting of a video captured by an UAV in a stationary way. The proposed approach using warping transformation based on the estimation of camera motion and a motion error. This work is an extension of the conference paper “Jitter-Free Registration for Unmanned Aerial Vehicle Videos” written by the authors and widely cited in the present document.
Of course, the paper is suitable for this journal. The paper is well organized: the paper is divided in six section and the titles of them are reasonable. The length of some sections is not reasonable, such as the Conclusions, Introduction and the Section 4: proposed approach.
The abstract is readable although the presence of a reference to the paper [1] strongly limits the comprehension of the topic paper and its motivations.
The introduction provides a sufficient description of the topic a deeper state of art should be reported. The section 3 is sufficient clear. The motivations for this study are clear, although in the paper lacks a precise description of some passages of the workflow that were hidden using a citation to the paper [1]. In order to obtain a uniform and accurate explanation of the proposed approach I strongly suggest writing the key passages of the previous work. Further details should be added in the section 5.
The conclusions are appropriated and coherent with the paper but can be extended with further considerations.
The number of references is sufficient.
General evaluations
The contribution is readable. The organization of the paper shall be improved, while the clarity is not always present. In some parts there are insufficient details, for this to be considered an accurate description of experimentation methods especially when the authors explain the implementation.
Of course, the paper is appropriate for this journal, the quality of writing is not always satisfactory: even if in general the paper is readable. In general, the length of the paper is not reasonable. The images are appropriate to the text and its contents but the position inside the document should be improved and some references inside the text should be added. The paper is technically accurate for the description of method, like the mathematical notation. The details in this paper are not so numerous, especially for the experimentation part where the authors talk about the implementation.
Further and detailed reviews are in the table below.
Review Table
Legend:
green = suggested review ; yellow = edit is no mandatory; cyan= please edit or add that is requested
|
Location |
Comments and review |
|
Page 1 , Rows 6-9 |
These sentences should be re-written in order to avoid references to the previous work |
|
Page 3, figure 1 |
Considerer to label the columns of images with “MFull” and “L2”. Moreover insert a reference to the section 5.1 to explain the |
|
Page 6, rows 138-139 |
The following sentence is not so clear, please rewrite. “We elaborate on this…” |
|
Page 6, rows 165-166 |
The following sentence is not so clear, please rewrite. “Hower important this minimization is…” |
|
Page 8, section 4.2 |
It’s not clear how did you perform the perspective transformation. Specifically, it’s not so clear how did you find the 4 points for the transformation? Did you use only 4 points or more than 4? If more did you use the Least Square Adjustment? Please insert further details into the section. |
|
Page 8, row 206 |
Please insert a reference for this sentence (“the tetragons defined by each set of point are convex”) |
|
Page 8, row 217 |
It is not so clear what are hb and ha, could you explain better? |
|
Page 9, row 230-231 |
Could you provide which type of acquisition was performed: 1080p or 1080i? If you used different type of acquisition for different acquisition please define them |
|
Page 9, row 237 |
Thank you for the information about the absences of gimbal for the CMOS camera, please provide the same information for the Go-Pro acquisition. |
|
Page 9, 256-259 |
How many points did you used? Have you perfomed the residual analysis? Could you explain in deep how did you use the RANSAC? |
|
Page 10,Section 5.4 |
Could you insert the trend of Frame Displacement indicator with a regression line for example. Please comment the results. |
|
Page 13, Section 6 |
The conclusions are too poor, please improve this section |
Author Response

(The authors gave the same response as above.)

Round 2
Reviewer 2 Report
Dear authors, thank you for your answers. First of all, I’d like to apologize to you for two mistakes:
- I did not provide the colors on the table;
- I lose during copy&paste operation part of sentence on the second row of the table
By the way, thank you for your answers and your work, the overall paper clarity was improved as well as the discussion of conclusion. I appreciated your modifications.
